# Impact of Residual Water Vapor on the Simultaneous Measurements of Trace $CH_4$ and $N_2O$ in Air with Cavity Ring-Down Spectroscopy

Qianhe Wei [1,2,3], Bincheng Li [1,*], Jing Wang [1], Binxing Zhao [1] and Ping Yang [2]

[1] School of Optoelectronic Science and Engineering, University of Electronic Science and Technology of China, Chengdu 610054, China; 202011050830@std.uestc.edu.cn (Q.W.); jingwang1230@uestc.edu.cn (J.W.); zhaobxing@uestc.edu.cn (B.Z.)

[2] Institute of Optics and Electronics, Chinese Academy of Sciences, Chengdu 610209, China; yangping@ioe.ac.cn

[3] University of Chinese Academy of Sciences, Beijing 100049, China

[*] Correspondence: bcli@uestc.edu.cn

**Abstract:** Methane ($CH_4$) and nitrous oxide ($N_2O$) are among the most important atmospheric greenhouse gases. A gas sensor based on a tunable 7.6 μm continuous-wave external-cavity mode-hop-free (EC-MHF) quantum cascade laser (from 1290 to 1350 cm$^{-1}$) cavity ring-down spectroscopy (CRDS) technique was developed for the simultaneous detection of $CH_4$ and $N_2O$ in ambient air with water vapor ($H_2O$) mostly removed via molecular sieve drying to minimize the impact of $H_2O$ on the simultaneous measurements. Still, due to the broad and strong absorption spectrum of $H_2O$ in the entire mid-infrared (mid-IR) spectral range, residual $H_2O$ in the dried ambient air due to incomplete drying and leakage, if not properly accounted for, could cause a significant influence on the measurement accuracy of the simultaneous $CH_4$ and $N_2O$ detection. In this paper, the impact of residual $H_2O$ on the simultaneous $CH_4$ and $N_2O$ measurements were analyzed by comparing the $CH_4$ and $N_2O$ concentrations determined from the measured spectrum in the spectral range from 1311 to 1312.1 cm$^{-1}$ via simultaneous $CH_4$ and $N_2O$ measurements and that determined from the measured spectrum in the spectral range from 1311 to 1313 cm$^{-1}$ via simultaneous $CH_4$, $N_2O$, and $H_2O$ measurements. The measured dependence of $CH_4$ and $N_2O$ concentration errors on the simultaneously determined $H_2O$ concentration indicated that the residual $H_2O$ caused an under-estimation of $CH_4$ concentration and over-estimation of $N_2O$ concentration. The $H_2O$ induced $CH_4$ and $N_2O$ concentration errors were approximately linearly proportional to the residual $H_2O$ concentration. For the measurement of air flowing at 3 L per min, the residual $H_2O$ concentration was stabilized to approximately 14 ppmv, and the corresponding $H_2O$ induced errors were −1.3 ppbv for $CH_4$ and 3.7 ppbv for $N_2O$, respectively.

**Keywords:** cavity ring-down spectroscopy; trace gas detection; residual water vapor; drying

## 1. Introduction

Global warming is considered to be the cause of climate change, and it has brought great consequences to human society [1]. Methane ($CH_4$) and nitrous oxide ($N_2O$) are two of the most important atmospheric greenhouse gases, contributing significantly to global warming as well as climate change. Their global warming potential (GWP) for a time horizon of 100 years are 25 [2] and 298 [3] times greater than $CO_2$. Due to their excellent stability and long atmospheric life periods, even small changes in $N_2O$ and $CH_4$ concentrations in air will have a long-term effect on the atmosphere [4]. Therefore, highly sensitive and precise measurements of $CH_4$ and $N_2O$ concentrations and their temporal variations in the atmosphere with ppbv (parts per billion by volume) accuracy are essential for environmental monitoring and greenhouse gas controlling [5,6]. Up to now,



plenty of techniques have been developed to detect trace gases, such as non-dispersive gas sensing including non-dispersive infra-red (NDIR) absorption spectroscopy, tunable diode laser absorption spectroscopy (TDLAS), cavity ring down spectroscopy (CRDS), and photoacoustic spectroscopy (PAS) [7,8]. Mahbub et al. have recently designed rapidly pulsed near-infrared light emitting diodes (NIR LED) based nondispersive infrared (NDIR) spectroscopy at 1.65 μm for continuous remote sensing of atmospheric methane ($CH_4$), and the limits of detection (LOD) of $CH_4$ is 300 ppm [9]. Shao et al. have used a single DFB diode laser emitting at 2.33 μm combined with the TDLAS technique for continuously detecting atmospheric CO and $CH_4$ at a level of 0.73 and 36 ppbv, respectively [10]. Among these various techniques, CRDS is a direct absorption technique with a significantly improved sensitivity than the conventional direct absorption spectroscopy due to its long effective absorption path length and insensitivity to intensity fluctuations of the light source [11–13], which has been widely used to detect trace gases in real time with ultrahigh sensitivity and relatively low system complexity in recent years [14]. With the rapid development of mid-infrared (mid-IR) quantum cascade lasers (QCL) and QCLs utilized in CRDS to employ the strongest absorption lines for trace gas detection, the sensitivity of CRDS was further improved [15–17] and has achieved the required sensitivities for real-time monitoring of trace species from ppmv (parts per million by volume) down to the pptv (parts per trillion by volume) levels. For example, Banik et al. utilized a 5.2 μm EC-QCL CRDS for the sensitive measurements of $N_2O$ at a level of 4.5 ppbv [18]. Maity et al. applied CRDS with an EC-QCL operating between 7.5 and 8 μm for detecting $CH_4$ at a minimum detection limit of 52 ppbv [19]. Tang et al. developed a CRDS setup for the simultaneous $CH_4$ and $N_2O$ detection of ambient air in the spectral range between 1290 and 1350 $cm^{-1}$ with the ppt-level sensitivity and ppb-level accuracy by drying the water vapor ($H_2O$) inside the sample cell down to sub-ppm level so the influence of $H_2O$ on the simultaneous $CH_4$ and $N_2O$ measurements becomes negligible [20].

However, the presence of gases other than the target species during the measurement will seriously affect the measurement accuracy [21–25]. Water vapor, one of the most important interference gases in the atmosphere, has a wide absorption spectrum from microwave to far-IR, and especially in the mid-IR spectral region. The spectrum of $H_2O$ cited from the HITRAN database [26] is shown in Figure 1a. Additionally, the concentration of $H_2O$ in the atmosphere varies in a large dynamic range from nearly 0 up to 4% over temporal and spatial scales [27] and mixing ratios of $CH_4$ and $N_2O$ are significantly affected by variations of water vapor. Moreover, without drying almost no CRD signals can be experimentally detected by the CRDS experimental setup used in this paper due to the high $H_2O$ concentration (in 1% level) in the ambient air and the limitation of the maximum detectable absorption coefficient (~$3 \times 10^{-5}$ $cm^{-1}$, corresponding ring-down time is about 1.5 μs) of the setup. According to the HITRAN database, the spectral absorptions for a typical ambient concentration of $CH_4$ (2 ppmv) and $N_2O$ (0.3 ppmv) with an $H_2O$ concentration of 10 ppmv and 1.4%, respectively are given in Figure 1b. Generally, $H_2O$ in the ambient air can be largely removed by passing the air over molecular sieves [28–30] and the concentration of $H_2O$ can be effectively decreased to even less than 1 ppmv [20]. In this case, its impact on the $CH_4$ and $N_2O$ detection becomes negligible. However, it is possible that if over-drying, the molecular sieve might also adsorb target gases and in turn affects the retrieval of the target quantity due to its non-uniformly distributed pore sizes. A more practical way may be reducing the amount of molecular sieve to minimize its influence on the target gas detection. In this case, the residual $H_2O$ in the levels from several ppmv up to 100 ppmv exists in the dried air due to incomplete drying. In addition, a possible leakage in the gas-handling system may also cause the increase of $H_2O$ concentration inside the sample cell.

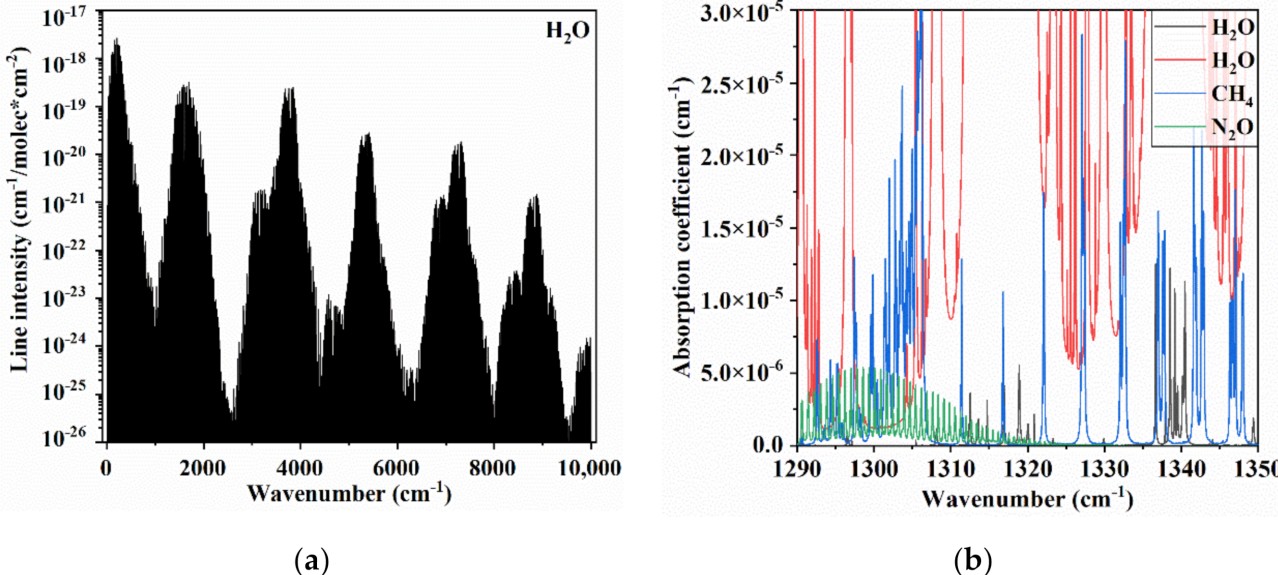

**Figure 1.** (**a**) The spectral lines of $H_2O$ from HITRAN2016 at 1 atm pressure and 296 K temperature. (**b**) The absorption coefficients of 2 ppmv $CH_4$, 0.3 ppmv $N_2O$, and 10 ppmv as well as 1.4% $H_2O$ at 1 atm pressure and 296 K temperature using HITRAN database data and Voigt linear simulation.

In this paper, a CW-CRDS experimental setup with a mode-hop-free (MHF) EC-QCL operating between 1290 to 1350 cm$^{-1}$ is applied to simultaneously detect $CH_4$ and $N_2O$ in clean laboratory air. In this experimental setup, the $H_2O$ is dried to a concentration from several ppmv to 200 ppmv by passing the measured air through molecular sieve filling tubes. A method is developed to determine simultaneously the concentrations of $CH_4$, $N_2O$, and residual $H_2O$ by measuring the absorption spectrum in a spectral range covering the absorption lines of $CH_4$, $N_2O$, and $H_2O$. The impact of residual $H_2O$ on the simultaneous $CH_4$ and $N_2O$ measurements is analyzed by comparing the results that with and without determining simultaneously the concentration of residual $H_2O$ is increased with time due to the slow leakage of the laboratory air into the closed ring-down cavity (sample cell) of the CRDS setup. Then, the same method is applied to analyze the impact of residual $H_2O$ on the simultaneous $CH_4$ and $N_2O$ measurements in flowing air.

## 2. Experiments

The apparatus used for the CRDS experiment is similar to that described in [20] and is schematically depicted in Figure 2. In brief, a tunable mid-IR external-cavity CW-MHF QCL (41074-MHF, Daylight Solutions, San Diego, California, USA) with an MHF tuning range of 1290–1350 cm$^{-1}$ and power nearly 160 mW is utilized as the light source. The QCL emits a collimated laser beam with a narrow spectral linewidth (<30 MHz or 0.001 cm$^{-1}$) which meets the requirements for high-precision spectral measurements. Moreover, the QCL is tuned via the laser controller with a step of 0.01 cm$^{-1}$ (with accuracy < 0.003 cm$^{-1}$). In order to avoid back reflections of light from other optical components affecting the laser output, an optical isolator with central wavelength 7.2 μm and isolation ratio >30 dB (FIO-5-7.2, Innpho, Verona, NJ, USA) is placed in front of the laser. Then, the light passes through an acousto-optic modulator (AOM) (I-M041, Gooch and Housego, Ilminster, UK) acting as a fast-optical switch that produces the zeroth and first-order diffraction beams. The AOM is controlled by a high-speed (with response time < 50 ns) threshold trigger circuit. The first-order diffraction beam is coupled into the stable high-finesse ring-down cavity which consists of a 50 cm long quartz-coated sample cell and two high-reflectivity plane-concave mirrors (CRD Optics) with reflectivity R > 99.98% and 1 m radius of curvature attached to two ends of the ring-down cell. Three piezoelectric transducers (PZT, Model PE-4, Thorlabs, Newton, NJ, USA) mounted on the rear cavity mirror is applied to achieve the periodic laser-cavity coupling. A triangular signal (Vpp = 5 V, frequency = 40 Hz)

is applied to the PZTs to modulate the cavity length over one half of the wavelength. A periodic resonant signal ($TEM_{00}$ mode) is built-up periodically inside the cavity due to mode-matching between the laser and cavity. The QCL beam leaking from the cavity is focused by a focusing lens and subsequently detected by a TE-cooled HgCdTe infrared photovoltaic detector (PVMI-4TE-8, Vigo, Poland). When the measured light intensity reaches a preset voltage value, the threshold trigger circuit sends a trigger signal to the AOM to switch-off the first-order beam. The ring-down signal is then recorded by a data acquisition (DAQ) card (M2i.3010, Spectrum Instrumentation, Großhansdorf, Germany) and analyzed by a MATLAB program. A vacuum pump (MPC 301Z, Welch, Concord, MA, USA), a pressure gauge (LEX1, Keller, Winterthur, Switzerland), and a mass flow meter (EL-FLOW Select, Bronkhorst, The Netherlands) are used to control the pressure and gas flows inside the sample cell. Three plexiglass drying tubes filled with 3A molecular sieves (Rhawn, Shanghai, China) which can absorb molecules with a dynamic diameter less than 0.3 nm are connected to the gas inlet of the sample cell to remove the water vapor from the measured gas mixture flowing into the sample cell. It is experimentally found that only when the concentration of water vapor in the measured air is below a certain level, the CRDS signal becomes observable and measurable. Differed from that in [20], no 3A molecular sieves are put inside the sample cell for further drying in order to minimize the possible adsorptions of $CH_4$ and $N_2O$ by the desiccants. In this case, the $H_2O$ concentration inside the closed sample cell increases with time as the outside $H_2O$ leak into the sample cell due to the imperfect seal of the sample cell and the large $H_2O$ concentration difference inside and outside the sample cell. It is worth mentioning that the $H_2O$ leakage may also cause small variations of the $CH_4$ and $N_2O$ concentrations inside the sample cell as $CH_4$ and $N_2O$ concentrations in the laboratory room fluctuate with time during the measurement period.

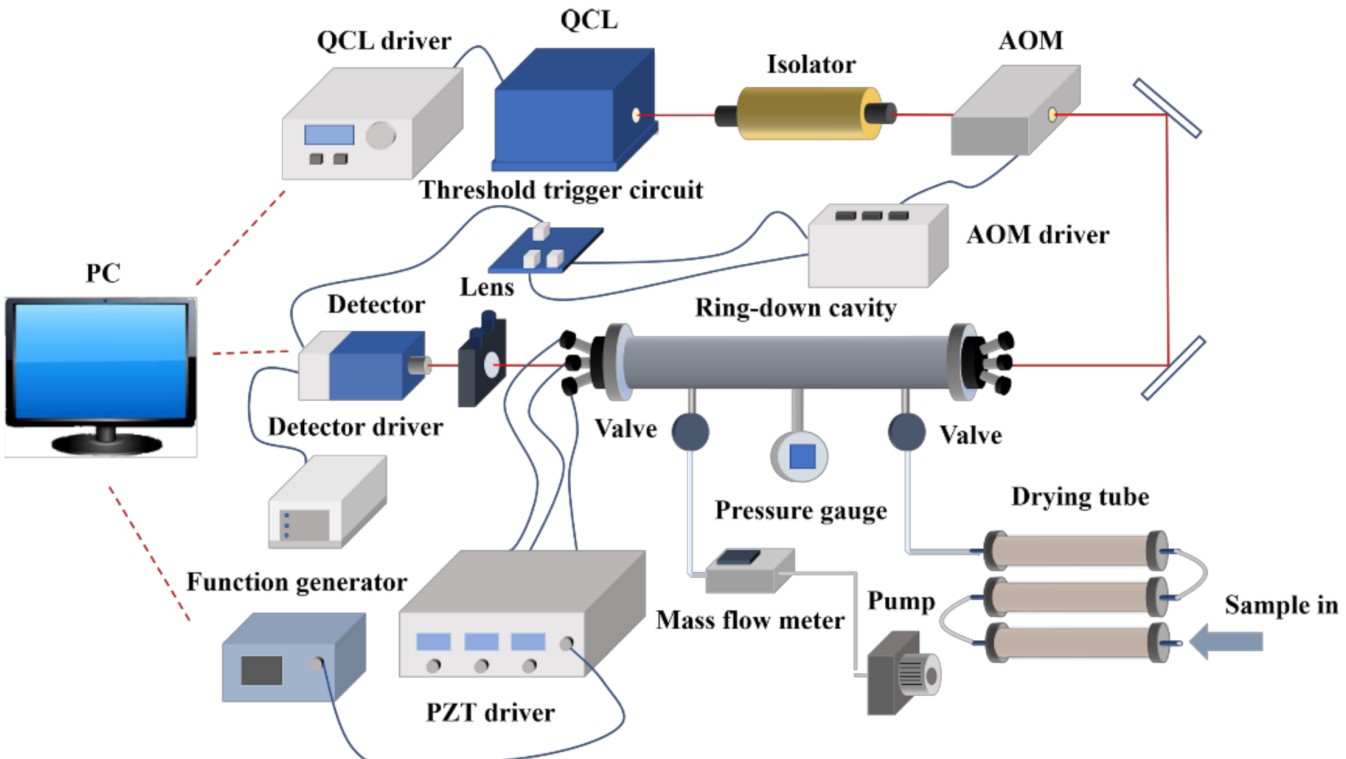

**Figure 2.** Schematic diagram of the cavity ring-down spectroscopy (CRDS) experimental setup.

Experimentally by determining the ring-down time $\tau$ from the measured ring-down signal, the wavelength-dependent absorption coefficient $\alpha(\lambda)$ of the target gas can be obtained by

$$\alpha(\lambda) = \frac{1}{c}\left(\frac{1}{\tau} - \frac{1}{\tau_0}\right) \tag{1}$$

where $c$ is the speed of light, $\lambda$ is the wavelength, and $\tau_0$ is the ring-down time in the empty sample cell (in our case, the "empty" sample cell is filled with high-purity nitrogen which has negligible absorption in the measurement spectral range). The absolute concentration of the target gas can be determined by comparing the measured absorption spectrum to the HITRAN database. For the simultaneous determination of multiple gases, the measured absorption spectrum can be expressed as

$$\alpha(\lambda) = \sum_{i=1}^{N} c_i \times \alpha_i(\lambda) + background \tag{2}$$

where $c_i$ is the concentration of gas $i$ ($i = 1, 2, \ldots, N$) in the gas mixture, and $\alpha_i(\lambda)$ is the corresponding absorption spectrum, which can be obtained from the HITRAN database. Once the number $N$ is determined, the concentrations of the multiple target gases can be determined by multi-parameter fitting the measured absorption spectrum to Equation (2). The absorption of interference gases can be treated as the background term which is set as a free parameter in the multi-parameter fitting. From Equation (2), to minimize the influence of background absorption on the simultaneous determination of multiple target gases, the level of background absorption should be as low as possible compared to the absorption caused by the target gases. In this case, the selection of wavelength range for the absorption spectrum measurement of multiple target gases is essential.

The gas mixture used in the experiment is the ambient air collected from the laboratory room with $H_2O$ partially removed by molecular sieves. The relative moisture level in the laboratory room fluctuates between 50% and 60%, which represents a $H_2O$ concentration range of 1.4–1.7%. The ambient air enters the sample cell (the ring-down cavity) through a vacuum valve (MPC 301Z, Welch). All the experimental data are recorded in the clean-room laboratory with approximately 296 K temperature and 950 mbar pressure (or 1 atmospheric pressure), respectively. For the simultaneous determination of $CH_4$ and $N_2O$, $N$ is set to 2 in Equation (2), and the impact of $H_2O$ is included in the background term. The concentration of $H_2O$ can also be simultaneously determined by setting $N = 3$ in Equation (2) and $H_2O$ being treated as an additional target gas when performing the multi-parameter fitting.

### 3. Results and Discussion

#### 3.1. Spectral Range Selection

To achieve simultaneous measurements of $CH_4$, $N_2O$, as well as $H_2O$, it is important to carefully select the measured spectral range. There are three factors that should be considered for wavelength selection: (1) Absorption lines of $CH_4$, $N_2O$, and $H_2O$ should all appear in the selected spectral range; (2) the intensity of absorption lines must be moderate for measurements of either strong absorptions of species present in trace amounts or measurements of weak absorptions of abundant species; (3) the absorption lines should be relatively isolated from each other and also from other $CH_4$, $N_2O$, and $H_2O$ lines to minimize the influence of neighboring lines and line-mixing effects. Figure 3a shows the absorption lines of $CH_4$, $N_2O$, and $H_2O$ in the MHF tuning range (1290–1350 cm$^{-1}$) of the QCL used in the experiment. From Figure 3a, the absorption lines in the spectral range from 1292 to 1308 cm$^{-1}$ are too abundant to be separated for both $CH_4$ and $N_2O$ measurements. On the other hand, the intensities of $N_2O$ absorption lines are too low in the spectral range from 1316 to 1348.8 cm$^{-1}$, which are less than $1.278 \times 10^{-19}$ cm$^2$/molecule. As a result, the spectral range from 1311 to 1313 cm$^{-1}$ is an excellent choice for the simultaneous measurements of $CH_4$, $N_2O$, and $H_2O$. The spectral lines of $CH_4$, $N_2O$, and $H_2O$ from 1311 to 1313 cm$^{-1}$ are shown in Figure 3b. In this spectral range, the absorption cross sections of $CH_4$, $N_2O$, and $H_2O$ are up to $2.587 \times 10^{-19}$, $3.079 \times 10^{-19}$,

and $1.459 \times 10^{-20}$ cm$^2$/molecule, respectively. For the simultaneous determination of CH$_4$ and N$_2$O, only the spectrum measured from 1311 to 1312.1 cm$^{-1}$ is used, in order to minimize the impact of H$_2$O on the CH$_4$ and N$_2$O measurements. On the other hand, the whole measured spectrum from 1311 to 1313 cm$^{-1}$ is employed when the concentration of H$_2$O is also simultaneously determined together with CH$_4$ and N$_2$O. By comparing the results obtained without and with the simultaneous determination of H$_2$O, the impact of H$_2$O on the simultaneous CH$_4$ and N$_2$O measurements can be analyzed.

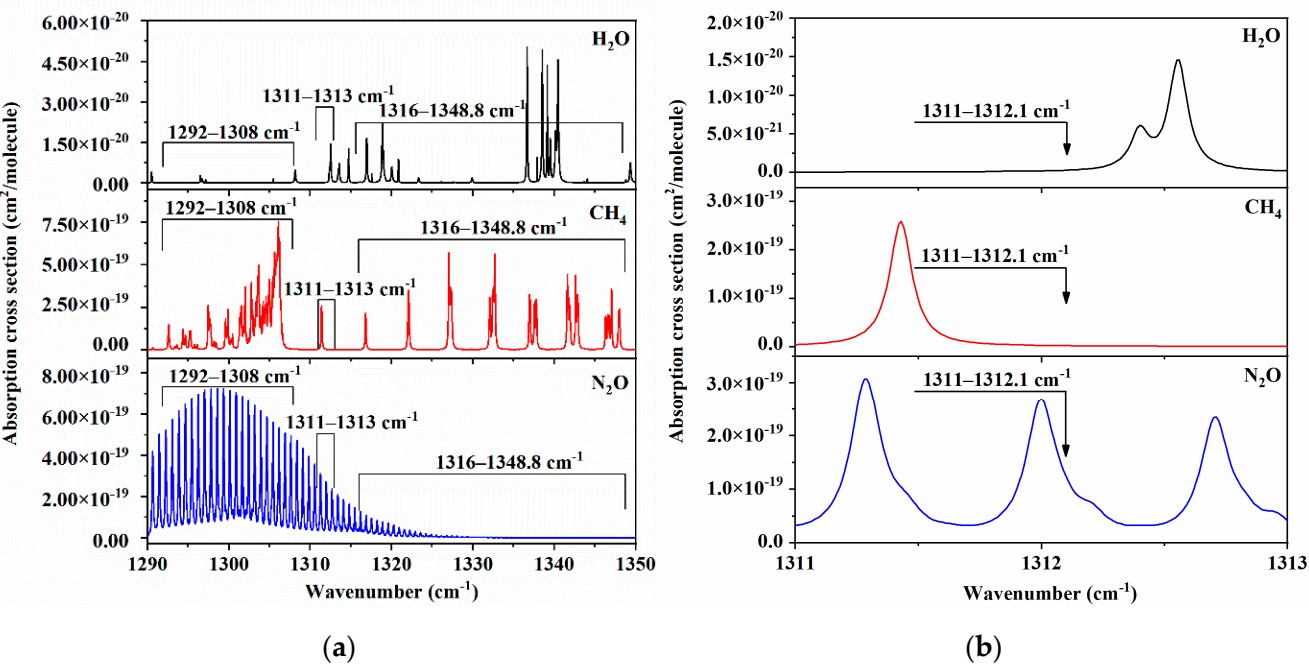

**Figure 3.** (**a**) The spectra of CH$_4$, N$_2$O, and H$_2$O from 1290 to 1350 cm$^{-1}$. (**b**) The spectra of CH$_4$, N$_2$O, and H$_2$O from 1311 to 1313 cm$^{-1}$.

### 3.2. Impact of Residual Water Vapor on CH$_4$ and N$_2$O Measurements

Firstly, the gas inlet and outlet valves connecting to the sample cell are open and laboratory air is fed into the sample cell by the vacuum pump at a flow rate of 3 L per min for 5 min. The H$_2$O in the sample cell is partially dried due to the molecular sieves inside the plexiglass drying tubes. Then, the gas inlet and outlet valves are adjusted to make the air pressure in the sample cell approximately equal to the outside air pressure (about 950 mbar), and then the valves are closed to form a closed sample cell filling with ambient air with H$_2$O partially removed. The experiment is performed to continuously measure the laboratory air for about 7 h from 13:30 to 20:30 on 30 July 2019. The QCL is tuned from 1311 to 1313 cm$^{-1}$ in 20 min and the measurement is repeated every 50 min (repeated 8 times in 7 h). The concentrations of CH$_4$, N$_2$O, and H$_2$O are then determined via fitting the measured spectra to Equation (2) with $N = 3$. The measured H$_2$O concentration is presented in Figure 4. An approximately linear relationship between the H$_2$O concentration and the measurement time is observed. This linear increase of H$_2$O concentration inside the sample cell with time is due to leakage of the sample cell and the gas system. From the linear dependence, the leak rate of H$_2$O can be determined to be approximately 27 ppmv/h. On the other hand, the intercept of 15 ppmv indicates the drying capacity of the molecular sieves at the flow rate of 3 L per min.

On the other hand, the CH$_4$ and N$_2$O concentrations can also be determined by employing the spectrum measured in the spectral range from 1311 to 1312.1 cm$^{-1}$. In this case, the concentration of H$_2$O is set to zero and its influence on the CH$_4$ and N$_2$O measurements is included in the background term in Equation (2). In the spectral range from 1311 to 1312.1 cm$^{-1}$, there are two N$_2$O absorption lines and one CH$_4$ absorption

line, which are used to determine the $CH_4$ and $N_2O$ concentrations. As the $CH_4$ and $N_2O$ lines are somewhat away from the $H_2O$, absorption lines appeared at approximately 1312.5 cm$^{-1}$ so the influence of $H_2O$ is minimized. Still, the influence of $H_2O$ might be not negligible due to the wide broadening of the $H_2O$ absorption lines. To analyze the impact of the residual $H_2O$ on the simultaneous $CH_4$ and $N_2O$ measurements, the $CH_4$ and $N_2O$ concentrations determined by the spectra measured from 1311 to 1312.1 cm$^{-1}$ and by that from 1311 to 1313 cm$^{-1}$ are compared, and the results are presented in Figure 5. Clearly, the existence of residual $H_2O$ has a significant influence on the $CH_4$ and $N_2O$ measurements. The impact on the $CH_4$ measurement is negative, indicating that the $CH_4$ concentration is under-estimated due to the existence of $H_2O$. On the other hand, the impact on the $N_2O$ measurement is positive, indicating that the $N_2O$ concentration is over-estimated due to $H_2O$. The $H_2O$ induced differences for both $CH_4$ and $N_2O$ concentrations increase with the increasing $H_2O$ concentration. Approximately, linear dependences of the $CH_4$ and $N_2O$ concentration differences ($H_2O$ induced measurement errors) on the $H_2O$ concentration are observed with slopes of approximately $-3.66 \times 10^{-4}$ and $+5.41 \times 10^{-4}$, respectively. That means, a 10 ppmv residual $H_2O$ will cause measurement errors of $-3.66$ ppbv for the $CH_4$ concentration and $+5.41$ ppbv for the $N_2O$ concentration. Or in other words, for a stable air measurement, the concentration of the residual $H_2O$ inside the sample cell must be below approximately 1.85 ppmv in order to keep the $H_2O$ induced measurement errors for the $CH_4$ and $N_2O$ measurements below 1 ppbv.

In Figure 6, the first and eighth measured and fitted spectra from 1311 to 1313 cm$^{-1}$ and the corresponding residuals are shown. In the eighth measurement, the peak absorption of $H_2O$ is significantly higher than that of $N_2O$ and $CH_4$ and many data points around the $H_2O$ peaks become undetectable due to the reduced signal-to-noise ratio (SNR) of the ring-down signals. It is clearly indicated that the fitting residual of $N_2O$ is larger than $CH_4$, since the second absorption peak of $N_2O$ is closer to the absorption peak of $H_2O$ and is more affected by the changes of $H_2O$ concentration.

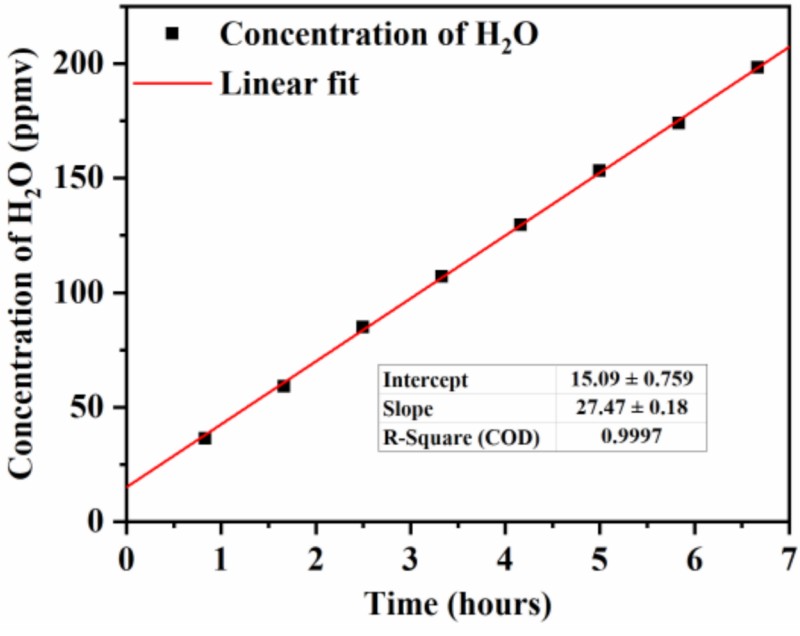

**Figure 4.** The linear regression of $H_2O$ concentration.

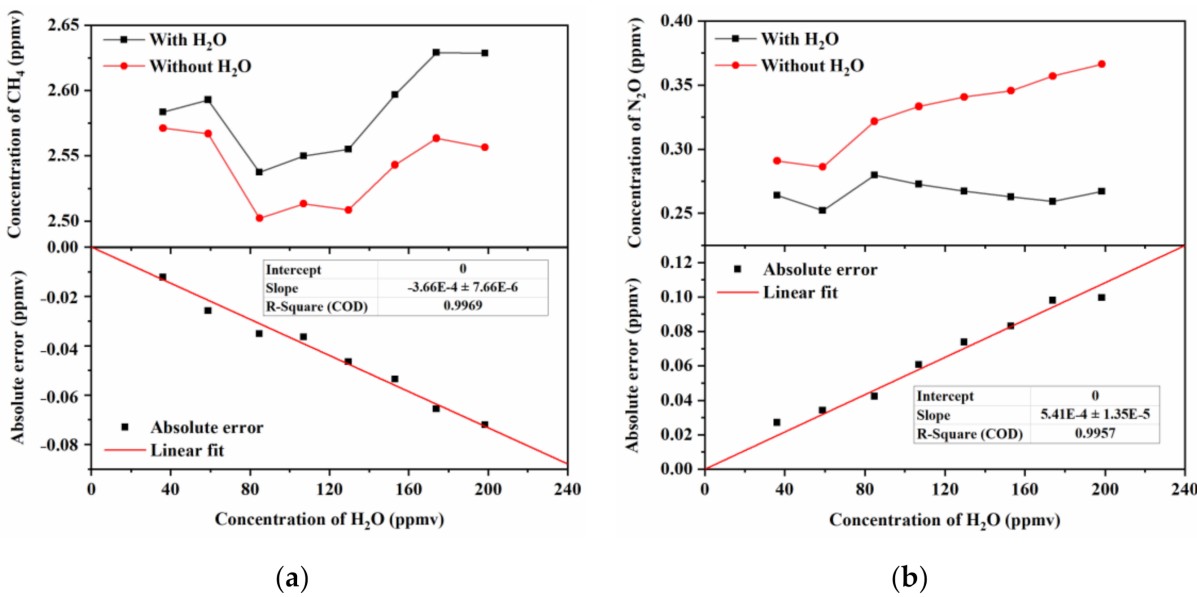

(a)                                                                  (b)

**Figure 5.** The differences of (**a**) $CH_4$ and (**b**) $N_2O$ concentrations between that determined with and without simultaneously determining the $H_2O$ concentration.

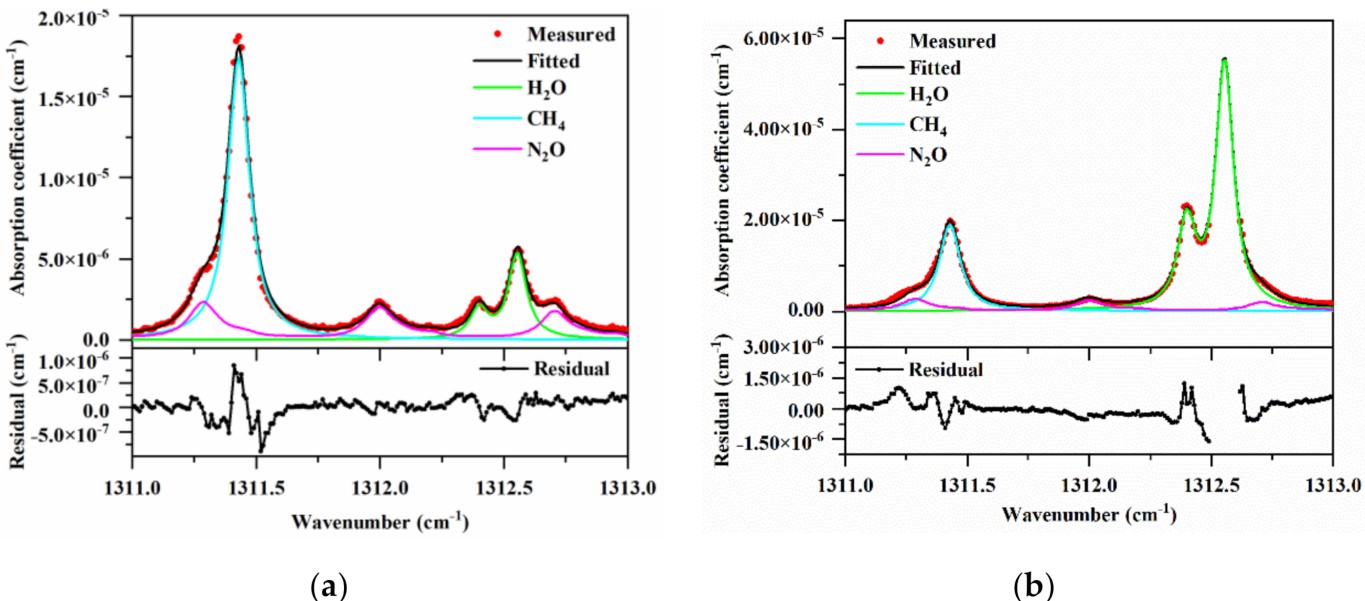

(a)                                                                  (b)

**Figure 6.** (**a**) The first and (**b**) eighth measured and fitted spectra and the corresponding residuals.

### 3.3. CH₄ and N₂O Measurements of Flowing Air

In practical applications, the $CH_4$ and $N_2O$ concentrations in the ambient air need to be monitored continuously. In this case, the measured air is flowing through the sample cell. Therefore, the simultaneous measurement of $CH_4$ and $N_2O$ concentrations is performed with flowing air and the impact of residual $H_2O$ on the simultaneous $CH_4$ and $N_2O$ determination is analyzed. This experiment is performed by keeping the sample cell open and flowing the laboratory air through the sample cell continuously at a flow rate of 3 L per min for 7 h from 8:30 to 15:30 on 1 August, 2019, and the measurements are performed with the same procedure as in Section 3.2. The results are presented in Figure 7. Obviously, the $H_2O$ concentration initially shows a downward trend in the first 3 h, and then gradually becomes stabilized to around 13–15 ppmv, which is consistent with the value (15 ppmv)

obtained in Section 3.2 showing the drying capacity of the molecular sieves at the flow rate of 3 L per min. Due to the impact of residual $H_2O$, the $CH_4$ and $N_2O$ concentrations are under-estimated by approximately $-1.3$ ppbv and over-estimated by approximately 3.7 ppbv, respectively. These $H_2O$ induced measurement errors are somewhat smaller than that estimated with the stable air in the closed sample cell (Section 3.2). From the measurement results with the flowing air, the residual $H_2O$ induced errors can be corrected once the $H_2O$ concentration becomes constant. Therefore, in real applications, once the $H_2O$ concentration becomes stabilized, there is no need to measure the residual $H_2O$ concentration repeatedly to reduce the measurement time (the time for one measurement is reduced by nearly half from 20 min to nearly 10 min in our case). From the results presented in Figures 5 and 7, the fluctuation of the $CH_4$ concentration during the whole measurement period is larger than that of $N_2O$, which is in agreement with our previous measurement results [20].

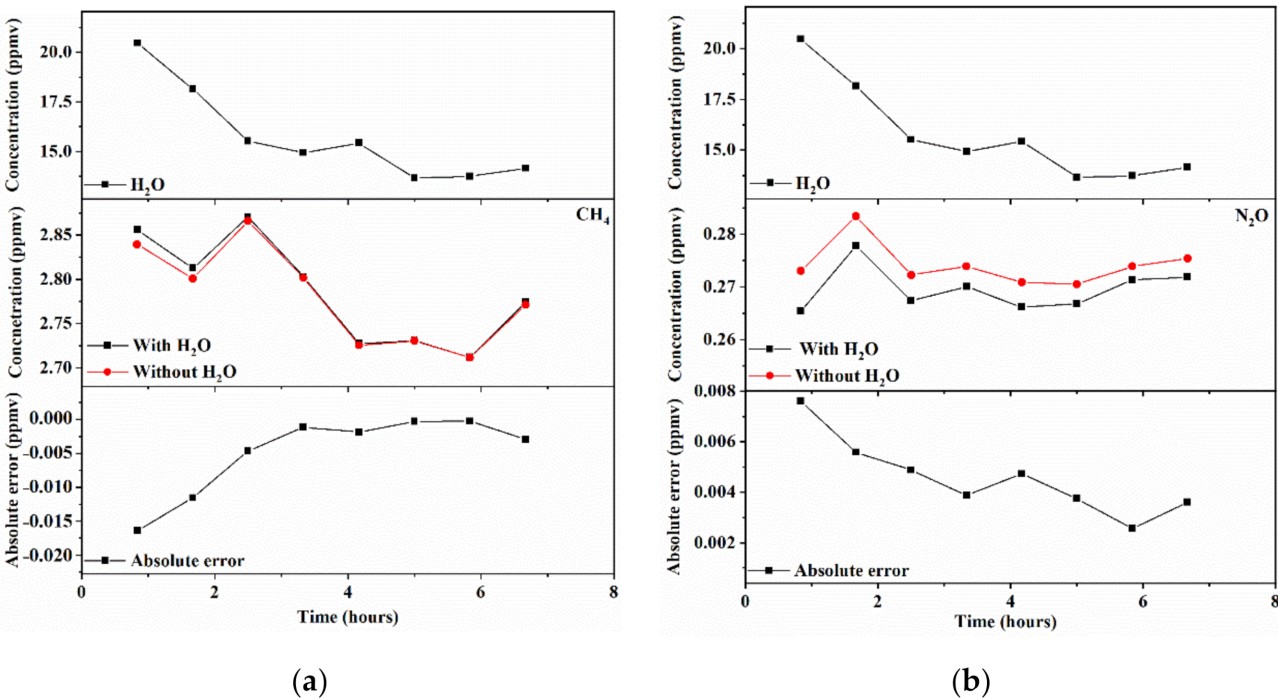

(**a**)                                                                                                         (**b**)

**Figure 7.** The differences of (**a**) $CH_4$ and (**b**) $N_2O$ concentrations between that determined with and without simultaneously determining the $H_2O$ concentration for flowing air.

## 4. Discussion

In this paper, the influence of residual $H_2O$ on the simultaneous measurement of $CH_4$ and $N_2O$ trace greenhouse gases in ambient air with tunable mid-IR CRDS coupled with EC-QCL in the spectral range from 1311 to 1313 cm$^{-1}$ has been investigated by treating the residual $H_2O$ either as a background gas or as a target gas, whose concentration is being determined simultaneously with the $CH_4$ and $N_2O$ concentrations. The experimental results showed that $H_2O$ has to be dried to a concentration below at least 200 ppmv. Therefore, the CRDS measurement is possible. Even so the residual $H_2O$ still caused a significant impact on the $CH_4$ and $N_2O$ measurements. The residual $H_2O$ caused an under-estimation of $CH_4$ concentration and over-estimation of $N_2O$ concentration, and the $H_2O$ induced measurement errors were approximately linearly proportional to the residual $H_2O$ concentration. The impact of residual $H_2O$ on the simultaneous $CH_4$ and $N_2O$ measurements could be minimized by measuring the spectra also including the absorption lines of $H_2O$ and determining the $H_2O$ concentration simultaneously. In practical applications, due to the linear dependence of the induced errors in $CH_4$ and $N_2O$ measurements on the residual $H_2O$ concentration, the influence of residual $H_2O$ could also be corrected if the residual

$H_2O$ concentration was stable and accurately measured. The results presented in this paper could be helpful to the more accurate monitoring of $CH_4$ and $N_2O$ in atmospheric air.

**Author Contributions:** Conceptualization, Q.W. and B.L.; formal analysis, Q.W. and B.L.; investigation designed, Q.W. and J.W.; resources, B.L., J.W. and B.Z.; software, Q.W. and J.W.; supervision, B.L., B.Z. and P.Y.; writing—original draft, Q.W.; writing—review and editing, B.L. All authors have read and agreed to the published version of the manuscript.

**Funding:** This research received no external funding.

**Institutional Review Board Statement:** Not applicable.

**Informed Consent Statement:** Not applicable.

**Data Availability Statement:** Not applicable.

**Conflicts of Interest:** The authors declare no conflict of interest.

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
