# Peer review of "Impact of Residual Water Vapor on the Simultaneous Measurements of Trace CH4 and N2O in Air with Cavity Ring-Down Spectroscopy"

_atmosphere, doi:10.3390/atmos12020221_

Round 1
Reviewer 1 Report
The article is very interesting and discussed the impact of residual H2O on the simultaneous CH4 and N2O measurements was analyzed using the cavity ringdown spectroscopy. The article provided sufficient introduction and also correctly described the methods. The results are presented in a good way by visualizing the data. However the article writing structure, English language, sentence structure need to be improved to understand it clearly. Other than that simple chemical formulas (format) are not written well, they need to be improved (see below). Hence I recommend the article to publish after a minor revision.
1.Please correct the Methane (CH4), nitrous oxide (N2O), water H2O, CO2 formulas through out the manuscript including title, abstract and in the manuscript.
2. Correct cm-1 to cm-1
3. Line 19-23: "In this paper, the impact of residual H2O on the simultaneous CH4 and N2O measurements was analyzed by comparing the CH4 and N2O concentrations determined from the measured spectrum in the spectral range from 1311 to 1312.1 cm-1 via simultaneous CH4 and N2O measurements and that determined from the measured spectrum in the spectral range from 1311 to 1313 cm-1 via simultaneous CH4, N2O and H2O measurements". - correct the chemical formulas and correct "was analyzed by" to were analyzed by.
4. Figure 1b: Why did you measured two different experiments with H2O; and the H2O peaks looks out of the range.
Author Response
Point 1.Please correct the Methane (CH4), nitrous oxide (N2O), water H2O, CO2 formulas through out the manuscript including title, abstract and in the manuscript.
Response 1: The chemical formulas(CH4, N2O and H2O) from Line 11-30, 34-93, 274-295 and 332-380 were corrected.
Point 2. Correct cm-1 to cm-1
Response 2: The cm-1 from Line 13, 22, 23 and 80 were corrected to cm-1.
Point 4. Figure 1b: Why did you measured two different experiments with H2O; and the H2O peaks looks out of the range.
Response 4: Firstly, we want to compare the absorption coefficient of water vapor in the two cases to further illustrate the importance of drying and analyzing the influence of water vapor on gas concentration measurement. Secondly, because the limitation of the maximum detectable absorption coefficient of this setup is 3×10-5 cm-1, we set the maximum ordinate value of Figure 1b to this 3×10-5 cm-1.

Reviewer 2 Report
The aim of the article is to present a research study regarding developing of one gas sensor based on a tunable 7.6 μm continuous-wave external-cavity mode-12 hop-free (EC-MHF) quantum cascade laser cavity ring-down spectroscopy (CRDS) technique for the simultaneous detection of CH4 and N2O in ambient air 14 with water vapor (H2O).
The topic of the paper is very important due to the fact that methane and nitrous oxide (N2O) are two of the most important atmospheric greenhouse gases with high impact on global warming as well as climate change. Consequently, there is a need to develop new highly sensitive and precise measurement techniques of CH4 and N2O concentrations in the presence of other air components for environmental monitoring and greenhouse gas controlling.
The topic is suitable for the journal. It is of broad international interest.
The structure of the article fulfills the structure of a research article. The paper is properly organized.
The paper’s title is brief and reflects the theme of the paper.
The abstract is informative and completely self-explanatory, briefly presents the topic, states the scope of the review, and points out major findings and conclusions.
The introductory part provides sufficient background information for readers in the immediate field to understand the problem that this study addresses. It clearly states the problem being investigated. The purpose and objectives are clearly stated. The novelty of the work is specified in the section.
The authors present in the Experiments part the equipment used for the CRDS experiment, the measurement protocol.
The authors accurately explain how experimental data were collected. The article identifies the procedures followed. They are ordered in a meaningful way. The methods and protocols are described in sufficient detail in order to allow another researcher to reproduce the results.
The following sections are suitable for the topic of the paper.
In the Results and Discussion part, the authors present and interpret the results of the experiments. They present the spectral range selection, the impact of the residual H2O vapor on CH4 and N2O measurements, the CH4 and N2O measurements on flowing air.
The authors should relate their findings to other researches results.
The paper ends with a section regarding conclusions. In this section the authors mentioned the major and specific conclusions of their research study. The quality of data’s interpretation and conclusions is suitable.
The abbreviations and nomenclature are used according to the applicable international standards and rules. The indexes are written correctly.
The figures are numbered sequentially, and they are clearly labeled and positioned close to the relevant text. Titles of figures are brief and informative. All the figures included are referred.
The references are accurate and relevant for the subject of the paper.
The English language is appropriate and understandable.
The used measurement units is according to the International System. The length of the paper is proper in keeping with its importance.
Taking into account all that I already mentioned I suggest publishing the paper in Atmosphere with minor changes.
Author Response
1.The chemical formulas(CH4, N2O and H2O) from Line 11-30, 34-93, 274-295 and 332-380 were corrected.
2.The cm-1 from Line 13, 22, 23 and 80 were corrected to cm-1.
3.Line 20 "was analyzed by" were changed to "were analyzed by".
